# Health Benefits of Coffee Consumption for Cancer and Other Diseases and Mechanisms of Action

**DOI:** 10.3390/ijms24032706

**Published:** 2023-01-31

**Authors:** Stephen Safe, Jainish Kothari, Amanuel Hailemariam, Srijana Upadhyay, Laurie A. Davidson, Robert S. Chapkin

**Affiliations:** 1Department of Veterinary Physiology and Pharmacology, Texas A&M University, College Station, TX 77843, USA; 2Master of Biotechnology Program, Texas A&M University, College Station, TX 77843, USA; 3Program in Integrative Nutrition and Complex Diseases, Department of Nutrition, Texas A&M University, College Station, TX 77843, USA

**Keywords:** coffee, AH receptor, Nrf2, redox, health

## Abstract

Coffee is one of the most widely consumed beverages worldwide, and epidemiology studies associate higher coffee consumption with decreased rates of mortality and decreased rates of neurological and metabolic diseases, including Parkinson’s disease and type 2 diabetes. In addition, there is also evidence that higher coffee consumption is associated with lower rates of colon and rectal cancer, as well as breast, endometrial, and other cancers, although for some of these cancers, the results are conflicting. These studies reflect the chemopreventive effects of coffee; there is also evidence that coffee consumption may be therapeutic for some forms of breast and colon cancer, and this needs to be further investigated. The mechanisms associated with the chemopreventive or chemotherapeutic effects of over 1000 individual compounds in roasted coffee are complex and may vary with different diseases. Some of these mechanisms may be related to nuclear factor erythroid 2 (Nrf2)-regulated pathways that target oxidative stress or pathways that induce reactive oxygen species to kill diseased cells (primarily therapeutic). There is evidence for the involvement of receptors which include the aryl hydrocarbon receptor (AhR) and orphan nuclear receptor 4A1 (NR4A1), as well as contributions from epigenetic pathways and the gut microbiome. Further elucidation of the mechanisms will facilitate the potential future clinical applications of coffee extracts for treating cancer and other inflammatory diseases.

## 1. Introduction

Coffee is among the most widely consumed beverages in the world, and it is estimated that over two billion cups of coffee are consumed daily [1,2]. Coffee intake is highly variable with respect to different countries, ages, and sex, and there appears to be a continuing increase in consumption which parallels, in part, the increasing number of specialty coffee shops in many countries. Coffee intake is often associated with the stimulant caffeine, which is a major component of coffee, and the average caffeine intake in the United States is 135 milligrams per day, which is equivalent to about 1.5 cups per day. Many individuals consume up to 6 cups of coffee per day and much higher amounts of caffeine. Although roasted coffee beans and brewed coffee contain high levels of caffeine, there are several hundred individual phytochemical-derived compounds in coffee, and these include chlorogenic acid/lignans, alkaloids, polyphenolics, terpenoids, melanoidins, vitamins, and metals [3].

Figure 1 illustrates some examples of the compounds identified in coffee, and these include the flavonoid quercetin, chlorogenic acid, caffeine, the alkaloid norharman (β-carboline), and the terpenoid cafestrol. The health impacts of coffee consumption have been extensively investigated and are associated with lower all-cause mortality, diabetes mellitus, dementia, Parkinson’s disease, cardiovascular disease, and many types of cancer [2,3,4,5,6]. The effects of coffee on mortality and other diseases have been extensively investigated in many countries and in groups of individuals that are both “normal” or have specific health problems. The results of recent and past studies clearly show the overall health benefits of higher coffee consumption compared to lower consumption; however, there are also many studies that do not correlate and, in some cases, report conflicting results. The reasons for these differences in some cases may be the failure to examine the effects of sex dependency; however, many other potential unknown confounders may be involved and these need to be further investigated.

## 2. Coffee and Health Benefits: Non-Cancer

### 2.1. Mortality

A number of recent studies (2020–present) have demonstrated that the higher consumption of coffee is associated with decreased mortality in both men and women. For example, in the UK Biobank study [2], the coffee intake of 395,539 individuals was collected between 2006–2010, and their overall disease-specific mortality was followed through 2020 (a median follow-up of 11.8 years). High levels of coffee intake (≥4 cups/day) were inversely associated with mortality from 30 of 31 diseases, with HRs ranging from 0.61–0.94, and these inverse associations were more predominant in women vs. men [2]. Interestingly the association between a high vs. low consumption of coffee in this population and mortality from various diseases was dependent on multiple variables, which include the sex of the individual, specific diseases, regular vs. decaffeinated coffee, and consumption levels. Examples of male vs. female differences with respect to a high vs. low consumption of coffee were associated with female/male HR values of 0.73/1.0 (digestive disorders), diabetes mellitus (0.74/0.92), and gout (0.71/0.59). In this study, there were “2 distinct clusters of medical conditions affecting mainly the cardiometabolic and gastrointestinal systems” [2]. In contrast, other studies did not observe inverse associations between coffee consumption and decreased mortality from neurogenerative diseases. Other studies in Korea [4], the United States [7,8,9,10], an Asia cohort [11], and an adult Mediterranean population [12] also reported that the higher consumption of coffee is associated with decreased mortality. Thus, the overall effect of coffee on mortality is comparable to previous and ongoing studies on other groups, including Seventh Day Adventists and the consumption of a Mediterranean diet, where high intakes of vegetables are also associated with decreased mortality [13,14].

### 2.2. Cardiovascular Diseases (CVDs)

In the Biobank study noted above [2], there was a strong association between decreased mortality from cardiometabolic diseases and the higher consumption of coffee [6], and this was also observed in a recent study on the Biobank population [15]. In contrast, a meta-analysis of 32 prospective cohorts reported that various studies showed that higher coffee consumption was associated with increased, decreased, or no effect on CVD [16]. In the UK Biobank studies, the higher consumption of coffee was associated with higher total cholesterol and LDL-cholesterol levels, and this was highest in the expresso coffee drinkers [17]. A meta-analysis of studies showed that higher coffee consumption was associated with “an increased risk of CHD in men and a potentially decreased risk in women” [16], and an increased risk was also observed in another analysis of the UK Biobank population [18]. Another recent report indicated that, among individuals with grade 2–3 hypertension, the HR mortality values were 0.98 (<1 cup/day), 0.74 (1 cup/day), and 2.05 (≥2 cups/day) compared to noncoffee drinkers [19]. In contrast, it was also reported that the medium–high consumption of coffee (3–5 cups/day) had a beneficial or neutral impact on hypertension and blood pressure [20]. Thus, the relationship between the higher consumption of coffee and mortality from cardiovascular diseases is somewhat variable between studies, and the factors that modulate the impact of the effects of coffee on this disease need to be determined.

### 2.3. Neurological Diseases

The linkage between higher coffee consumption and decreased mortality for neurologic diseases such as dementia, Parkinson’s disease, and Alzheimer’s disease have been extensively investigated, and the results has been variable. It was recently confirmed that high coffee consumption is associated with decreased risks of neurological disorders, including dementia, stroke, and Parkinson’s disease [16,17,19,21,22,23]. These results were also observed in the UK Biobank prospective study [24]. As noted above, another report using the UK Biobank did not observe any correlation between coffee consumption and decreased mortality from neurodegenerative diseases [2]. A negative finding was the association between the early age of onset of Huntington’s disease with increased coffee consumption, and this outlier might be due to the strong genetic origins of this debilitating autosomal dominant disease [25]. However, there is evidence for the improvement of specific neurologic conditions with coffee consumption. For example, the moderate consumption of mocha coffee in an elderly population was associated with higher cognitive and mood status [26]; coffee consumption enhanced the age at onset of Parkinson’s disease in Ashkenazi Jewish patients [27]; the results of a meta-analysis showed that coffee consumption reduced the risk of overall stroke, hemorrhagic, and ischemic stroke [28]. Thus coffee–neurological disease interactions are variable in terms of mortality; however, there is evidence of protection from nonlethal neurological diseases that also needs to be considered and more fully investigated.

### 2.4. Metabolic Diseases including Diabetes 

The higher consumption of coffee is also associated with a decreased risk of metabolic diseases and type 2 diabetes [29,30,31,32,33,34]. An analysis of plasma biomarkers in nondrinkers vs. individuals consuming ≥4 cups/day showed that in the latter group, the changes in biomarkers were consistent with favorable outcomes [31]. For example, higher concentrations of sex hormone binding globulin (SHBG) (5.0%), total testosterone (7.3 and 5.3% in women and men, respectively), and total (9.3%) and high molecular weight (17.2%) adiponectin were increased in the coffee drinkers. In contrast, the group consuming high amounts of coffee exhibited lower levels of inflammatory markers, such as interleukin-6 (−8.1%), soluble tumor necrosis factor receptor (−5.8%), and C-reactive protein (−16.6%) [31]. These results were obtained from two large prospective studies: the Nurses Health Study and the Health Professionals Follow-up Study in the United States [31]. There is also evidence that coffee consumption interacts with other factors that modify the association between coffee and metabolic diseases. For example, in patients with rheumatoid arthritis, coffee intake was associated with lower metabolic syndrome scores [33]; increased caffeinated and noncaffeinated coffee intake protected against nonalcoholic fatty liver disease (NAFLD) severity in individuals with type 2 diabetes [35], and similar results were observed in a normal population [36]. The protective effects of coffee on metabolic disorders were also observed in individuals with a history of gestational diabetes [37], diabetic retinopathy [38], and hepatitis B viral infection [39]. These results are complemented by several recent studies showing that high coffee consumption is also associated with protection from inflammatory bowel disease and gut recovery in gynecological patients from surgery [40,41,42].

### 2.5. Sex-Dependent Effects of Coffee Consumption

The sex-dependent development of some diseases has been described and is thought to be due to multiple factors; there is some evidence for differences with respect to the effects of coffee on males and females [43]. The higher consumption of coffee decreased the prevalence of metabolic syndrome in an adult Taiwanese population, and the protective effects were more pronounced in women [44]. In a UK Biobank study [2], the overall effects of high coffee consumption among subgroups of diseases were higher for women than men and this was particularly evident for functional digestive disorders and diabetes mellitus, whereas men were more protected from gout [2]. A meta-analysis of the risk of coronary heart disease also showed that coffee consumption was associated with lower risks for coronary heart disease in women than in men [16]. A recent review summarized the sex-dependent differences in several neurological and psychiatric disorders and used estimated caffeine consumption data to compare the association between these disorders [43]. Caffeine is more effective in women than in men for improving depression and Parkinson’s disease, and caffeine enhances anxiety in men more than in women. In this brief introduction to the association between coffee and a decreased risk of some diseases, we have primarily used references published from 2020–the present, and a similar selection of more recent articles will be used to review the association of high coffee consumption to decreased risks of cancer.

## 3. Coffee and Cancer

Coffee consumption and a decreased risk of cancer have been extensively investigated, and multiple studies confirm the inverse association between high coffee consumption and a decreased cancer risk [6,7,11,12,45,46]. Reports on the effects of coffee consumption on the overall and specific cancer risks are extensive and sometimes contradictory, and this review will primarily focus on the results of recent studies (i.e., 2020–the present) unless data for a particular type of cancer have not been published recently.

### 3.1. Gastrointestinal Tract

#### 3.1.1. Liver Cancer

A recent comprehensive review summarizes the lack of association between coffee consumption and cancers of the different digestive organs, pointing to the powerful protective effect of coffee against the risk of hepatocellular carcinoma [47]. Two studies using the UK Biobank participants confirmed that the higher consumption of all coffee types (including decaffeinated) decreased the risk of chronic liver disease [48] and liver cancer [49] but no other digestive cancers [50]. Similar results were observed for liver cancer in a Japanese population [49] and meta-analysis of other studies [51].

#### 3.1.2. Colon and Rectal Cancer

Two recent studies have confirmed previous reports in which higher coffee consumption did not decrease the risk of colon cancer [52,53,54], whereas another report indicated that ≥2 cups/day of decaffeinated coffee lowered the risk of colon cancer [55]. Two additional papers reported that higher coffee consumption was associated with decreased risk of colon cancer [56,57]. Moreover, in both studies, the interaction between genetic polymorphisms was observed, including the variant aryl hydrocarbon (AhR) rs2066853 gene [52]. It was also reported that higher coffee consumption was associated with some decreased risks of rectal cancer.

#### 3.1.3. Other Gastrointestinal Cancers

There was no association between high coffee consumption and the risk for pancreatic cancer [58,59] or gastric cancer [54,60,61]. Similar results were observed for overall esophageal cancer [62,63]. However, in a European prospective cohort, there was a decreased risk of esophageal squamous cell carcinoma [63], and a decreased risk was observed for East-Asian but not European participants, whereas an analysis of the UK Biobank data suggested that coffee intake had increased the risk of esophageal cancer [45].

### 3.2. Genitourinary Cancers

The effects of coffee consumption on genitourinary cancers have not provided definitive associations in epidemiological studies, and research in this area is ongoing.

#### 3.2.1. Prostate and Bladder Cancer

Some studies show that there is no association between high coffee consumption and decreased risk of prostate cancer [64,65,66], while other reports show an inverse relationship [67,68,69,70,71]. A recent study of a large Japanese cohort showed no association between higher coffee consumption and prostate cancer risk [72]; clearly, these studies are inconclusive, and other factors may be involved. For bladder cancer, individual studies and meta-analyses do not show an association between coffee consumption and bladder cancer [73,74].

#### 3.2.2. Renal Cancer

A recent Mendelian randomization study did not observe a decreased risk of renal cancer with increased coffee consumption [75]. These results were in contrast to a meta-analysis of several cohorts [76] and another large study [77], where a 20% reduced risk was observed in patients with coffee intakes of ≥2 cups/day. These effects may be due, in part, to coffee constituents, such as kahweol and cafestrol which inhibit the growth and migration of renal cancer cells [78].

#### 3.2.3. Endocrine Cancers

The endocrine status of an organ modulates the effects of various exogenous substances, including diet and beverages, such as coffee, and this has been observed for breast and endometrial cancers.

#### 3.2.4. Breast Cancer

A recent review concluded that “there is no association between coffee intake and breast cancer risk or a slight protective effect even at the higher dosages” [79]. However, breast cancer is a highly complex disease and is observed in both pre and postmenopausal women, and these factors may modify the effects of coffee. An analysis of the data from the Women’s Health Initiative [80,81] did not show an association between high coffee intake and decreased risk of breast cancer, and similar results were observed with the Cancer Prevention Study-II Nutrition cohort [82]. In contrast, there is also evidence that higher coffee consumption decreases breast cancer risk in postmenopausal and European women [83,84], women expressing the minor allele of the bcl-2 gene haplotype [85], and other bcl-2 polymorphisms.

#### 3.2.5. Endometrial Cancers

A meta-analysis of several studies showed no association between coffee consumption and the risk for endometrial cancer, whereas another large study demonstrated that coffee consumption decreased the risk for endometrial cancer, particularly among women with a body mass of ≥25 kg/m^2^ [86,87].

#### 3.2.6. Other Cancers

Results from lung cancer studies showed that coffee consumption increased [88,89] or decreased [90] the risk; coffee consumption had no effect [91] or decreased [92,93] the risk for glioma and decreased the risk of head and neck cancers [94]. Maternal consumption of coffee was either not associated with or increased the risk of childhood cancers [95,96]. This is an area that needs to be further investigated since the effects of in utero exposures on the offspring are being studied for many other chemicals, including endocrine-disrupting compounds and phytochemicals.

## 4. Chemopreventive and Chemotherapeutic Effects of Coffee

The dietary consumption of foods, including coffee, and specific diets, such as the health-promoting Mediterranean diet, contain many of the same classes of phytochemicals and are associated with some of the same health benefits linked to non-cancer and cancer endpoints. These benefits are usually derived from the long-term consumption of specific foods and beverages and are chemopreventive; namely, they decrease the risks of developing a disease. Too few studies are designed to examine their chemotherapeutic activities, which are the effects observed after disease diagnosis. Soldato and coworkers investigated the effects of coffee consumption on breast cancer patient outcomes from years 1–4 after an initial diagnosis. Based on their patterns of coffee consumption, there was no association with the clinical outcomes [97]. In contrast, the postdiagnosis effects of high coffee consumption by women in the Nurses’ Health Study were associated with lower breast cancer-specific mortality [98]. Another report showed that in a cohort of patients being treated for advanced metastatic colon cancer, coffee consumption was associated with a decreased risk of subsequent cancer progression and death [99]. For example, patients “who consumed at least 4 cups of coffee per day have a multivariate HR for OS of 0.64 (95% CI, 0.46–0.87) and for PFS of 0.78 (95% CI, 0.59–1.05)”. Significant associations were noted for both caffeinated and decaffeinated coffee [99]. These results suggest potential therapeutic applications for coffee, which may be of value for treating multiple cancers. Thus, either coffee or coffee extracts should be more extensively evaluated as cancer therapeutics that can be used in combination with ongoing therapies to enhance the overall survival from this deadly disease. This should also be investigated in other cancers.

## 5. Mechanisms of Coffee-Mediated Anticancer Activities

This review of cancer studies has primarily focused on coffee consumption and its anticancer activities, which can be both chemopreventive (before cancer diagnosis) and chemotherapeutic (after cancer diagnosis). The subset of coffee compounds that are chemopreventive and chemotherapeutic will not necessarily be the same compounds, and for those compounds that are both chemopreventive and therapeutic, their mechanisms of action for these two responses may also differ. Chemopreventive mechanisms are difficult to establish in human and rodent models. However, it is assumed that the compounds that reduce the formation of oxidative stress and other stressors and decrease radical formation and inflammation play a role in disease prevention. These chemopreventive pathways are also associated with the Mediterranean diet [100,101], which is enriched in phytochemicals similar to those observed in coffee. Figure 1 illustrates examples of some of the major classes of phytochemicals in coffee and include caffeine, quercetin, chlorogenic acid, cafestrol and norharman (β-carboline).

### Activation of Nrf2 by Coffee

A recent study examined the activities of the phytochemicals in coffee and concluded that their overall effects were not sufficient to account for the required radical scavenging anti-inflammatory activity observed in human and laboratory animal studies. It was proposed that the activation of nuclear factor erythroid 2 (Nrf2) and its protective pathways (Figure 2) play a major role in mediating the beneficial health effects of coffee [102]. For example, coffee/coffee phytochemicals induce or activate Nrf2 in cells under some oxidative stress [102,103,104,105]. Among the important Nrf2-regulated genes associated with antioxidant activity are glutathione peroxidase haem oxygenase-1, glutathione reductases, superoxide dismutase, quinone oxidoreductase, and several thioreductases. Nrf2 exists as a cytosolic dimer with Keap2, and it also interacts with Cullin 3-based ubiquitin ligase and this complex maintains basal cytosolic levels of the Nrf2-Keap heterodimer. A combination of factors, including those that inhibit Keap-Nrf2 interactions or enhance Keap degradation, results in the nuclear uptake of Nrf2, which binds to small musculoaponeurotic fibrosarcoma (sMaF) protein to form the Nrf2-sMaF heterodimer, which then interacts with the cis-acting antioxidant response element (ARE) in target gene promoters to activate gene expression. This results in the activation of antioxidant genes (e.g., glutathione reductase, glutathione peroxidase), multiple redox family genes, (e.g., catalase, haem oxygenase (1)), anti-inflammatory genes (e.g., interleukins, interferons, tumor necrosis factor), drug metabolism enzymes (e.g., epoxide hydrolase, UDP–glucuronyl transferases, CYP1B1) and many other genes/pathways. All of these induced, Nrf2-dependent pathways/genes play diverse roles as cellular protective proteins. In addition, several reports demonstrate that the AhR and its ligands co-operatively enhance Nrf2 pathways [105,106,107,108]. One of these pathways involves the aryl hydrocarbon receptor (AhR)-dependent induction of CYP1A1 and CYP1A1-dependent substrate metabolism, which, in turn, enhances Nrf2 expression. The AhR also directly binds to the Nrf2 promoter and induces the levels of this protein [106].

The protective properties of Nrf2 are dependent on the induction of Nrf2 and Nrf2-dependent genes, as illustrated in Figure 2, and there is evidence that many of the constituents in coffee, including chlorogenic acids, phenolics, caffeine, cafestrol, and kahweol, are inducers of Nrf2 [109,110,111,112,113,114,115,116,117,118,119]. Moreover, a recent study showed that aqueous coffee extracts induced Nrf2 and Nrf2-dependent genes in various cell lines, and the co-operative role of the coffee-induced activation of AhR/Nrf2 had been reported in several in vitro and in vivo studies involving coffee extracts [120,121,122,123,124,125]. The activation of Nrf2 and Nrf2 protective genes by coffee and its individual components is consistent with their activity to protect nontransformed cells from oxidative stress. However, there is also evidence that the cytotoxicity of coffee components can occur by multiple pathways, including the disruption of the mitochondrial membrane, and this results in a loss of mitochondrial membrane potential and ROS-dependent cytotoxicity [125,126,127,128,129,130,131] (Figure 3). This pathway has been observed for many ROS-inducing phytochemicals and includes the inactivation of cMyc and cMyc-regulated microRNAs (miRs), the induction of ZBTB genes, which repress Sp1-, Sp2-, and Sp3-regulated genes/pathways [131]. Thus, coffee components activate cell context-dependent antioxidant and oxidative stress pathways that are primarily but not exclusively linked to chemopreventive and chemotherapeutic pathways in non-cancer and cancer cells, respectively.

Although the activation of Nrf2 plays an important role in alleviating oxidative stress, the increased levels of Nrf2 in cancer cells are linked to malignant progression and drug resistance [132]. Coffee extract-induced Nrf2 and the similar effects of other phytochemical-enriched diets may be beneficial in normal cells that require low levels of ROS; however, in cancer cells, this can result in enhanced carcinogenesis [132,133]. Nrf2 levels in cancer cells are enhanced due to overexpressed mutations that activate Nrf2 or mutations in Keap, and this enhances the pro-oncogenic pathways and the development of resistance to anticancer agents [133,134]. For example, in hepatic progenitor cells, Nrf2 induces malignant transformation due to the activation of the wnt-β-catenin pathway [135]; suppression of Nrf2 by histone lysine methyltransferase SETDB2 inhibits the progression of lung adenocarcinoma cells, whereas the decreased SETDB2 in these cells enhances Nrf2-mediated tumorigenesis [136]. It has also been reported that the chemically or genetic-mediated downregulation of Nrf2 increases ROS-medicated cell death and reverses some drug resistance [137,138,139]. Interestingly, coffee extracts contain trigonelline, which inhibits Nrf2 activity and thereby modulates the potentially beneficial and harmful effects of NRF2. The inhibitor of Nrf2 trigonelline and other coffee components inhibit cancer cell growth, migration, and drug resistance [140,141,142,143,144,145]. In contrast, Nrf2 inhibitors, such as trigonelline, can interfere with Nrf2-mediated antioxidant activity and thereby reverse some of the protective effects of Nrf2. Thus, coffee and some of its key chemical components (chlorogenic acid and phenolics trigonelline) and pathways (Nrf2 and ROS) exhibit cell context-dependent opposing effects. This suggests that other cell-specific factors must also contribute to the regulation of other pathways/genes that are associated with the health benefits of this beverage. Moreover, the issue of coffee and other phytochemicals and their beneficial effects need to be further evaluated with respect to cancer, where Nrf2 may not be beneficial.

## 6. Receptor-Mediated Responses

The thousands of membrane-associated and intracellular receptors are the key genes that are the sensors of dietary and intracellular cues and are required for maintaining cellular homeostasis and are involved in pathophysiology. PubMed lists over 40,000 manuscripts under the heading “Receptors and aging”, and these include papers on most of the age-related responses, such as decreased mortality and improved neurological responses, which are associated with the higher consumption of coffee. However, the direct linkages between the individual compounds in coffee extracts to the specific receptors that play a significant role in coffee extract-induced health benefits have been understudied. This review identifies some receptors that may serve as targets for coffee components and their associated beneficial health effects.

### 6.1. Aryl Hydrocarbon Receptor (AhR)

As indicated above, Nrf2 and the AhR are co-operatively activated by coffee extracts, and studies in our laboratories have primarily focused on the direct effects of coffee extracts as AhR ligands in the intestine (Figure 4A) [146]. AhR plays an important role in multiple organs/tissues, and there is extensive evidence that AhR is involved in aging and is related to diseases, suggesting that AhR and its ligands can affect these conditions [147,148,149,150,151]. Aqueous coffee extracts induced the Ah- responsive CYP1A1, CYP1B1, and UGT1A1 genes in Caco-2 colon cancer cells and YAMC mouse colonocytes, and these responses were abrogated in their corresponding AhR knockout (AhR-KO) cell lines. In addition, Ah responsiveness was primarily observed in a chloroform extract of aqueous coffee extracts, and chromatographic and cell culture analysis indicated that caffeine was not an AhR-active compound. The chloroform extracts from coffee were separated by this layer chromatography into three bands: a less polar band, a caffeine band, and a polar band. The extracts of the polar and less polar bands were AhR-active [146]. A comparison between aqueous extracts of unroasted and roasted ground coffee demonstrated that the high levels of AhR-active extracts in brewed coffee were primarily due to the roasting process. In vivo studies showed that coffee extracts inhibited DSS-induced colonic inflammation and inhibited the growth of organoids enriched in colonic stem cells. This latter response is consistent with anticancer activity since colonic stem cells are precursors for the development of colon tumors. We observed some of the same responses for the alkaloid norharman(β-carboline), which is one of the AhR agonists in coffee extracts that contributes to the AhR-dependent responses observed in the study. These results for coffee were observed in wild type but not AhR-KO cells and animal models and are consistent with previous studies showing a protective role for AhR and its ligands in colonic inflammation and cancer [152,153,154,155,156,157]. The identities of most AhR ligands in coffee extracts have not been determined; however, it should be pointed out that AhR and its ligands can be both beneficial and harmful, and these responses are organ/tissue-context-dependent [158]. The beneficial effects of AhR ligands in colon cancer are well supported; however, AhR is also a pro-oncogenic factor in other tumor types, including head and neck cancer [159]. Thus, at least some of the health-promoting effects of coffee are mediated through its binding and activation in AhR signaling.

### 6.2. Nuclear Receptor 4A1 (NR4A1, Nur77)

NR4A1, NR4A2 (Nurr1), and NR4A3 (Nor1) are orphan nuclear receptors and the immediate, early genes that respond to stress and play a role in maintaining cellular homeostasis and pathophysiology [160,161]. These receptors are often increased in the diseased cell types that are associated with stress, and this includes many solid tumors and derived cancer cell lines [162]. It was hypothesized that NR4A orphan nuclear receptors were the potential targets for anti-aging interventions [163] due to the functions of these receptors in the immune system and stress-related diseases, and this has now been confirmed by several studies [164,165,166,167]. For example, age-related cognitive impairment in mice results in decreased NR4A expression, and treatment with a bis-indole-derived NR4A2 ligand enhanced long-term spatial memory and rescued memory deficits in mouse models [166]. Age-related renal fibrosis was also suppressed by NR4A1 [165], and there are other reports showing that NR4A is involved in age-related diseases. Recent studies reported that polyphenolics, including flavonoids such as quercetin and kaempferol that occur in coffee, bind to NR4A1 in cancer cells, act as receptor antagonists, and inhibit pro-oncogenic NR4A1-regulated genes and pathways (Figure 4B) [168]. When using Rh30 rhabdomyosarcoma cells as a model, it was reported that quercetin and kaempferol decreased NR4A1-dependent transactivation, inhibited Rh30 cell growth, survival, and invasion, decreased tumor growth, and also decreased mTOR signaling and β1-integrin expression. All of these effects were also observed after NR4A1 knockdown or after treatment with other NR4A1 antagonists [168]. Ongoing studies have identified multiple polyphenols that are present in coffee extracts and in Mediterranean diets. Thus, it is likely that NR4A1 and its phytochemical ligands may play important roles in the health-promoting effects of coffee and phytochemical-enriched diets, and this needs to be further investigated.

## 7. Other Coffee-Induced Pathways

### 7.1. Changes in DNA Methylation

Based on the health benefits of coffee consumption, an epigenome-wide association meta-analysis of DNA methylation with coffee drinkers was reported [169]. The data were generated from epigenome-wide association studies of coffee (and tea) consumption among 15,789 European and African American participants from 15 cohorts. An analysis of the data identified several CpGs that were related to coffee consumption, and these included the AHRR, F2RL3, FLJ43663, HDAC4, GFL1, and PHGDH genes. The relationship between these genes and the effects of coffee is uncertain, although it was suggested that PHGDH might be associated with protection from liver damage. In addition, AHRR represses AhR activation and could also affect AhR activity. The nuclear receptor NR4A1 regulates the methyltransferase gene G9A [168], and, possibly, the other methyltransferases and the epigenetic effects of coffee could be due, in part, to NR4A1 ligands, such as flavonoids and other polyphenolics in coffee extracts. Thus, the linkage between coffee extract-induced epigenetic changes as causal factors needs to be further investigated, along with parallel comparisons with animal models.

### 7.2. Coffee-Induced Microbiome Changes and Health

The gut microbial populations and their metabolites strongly influence intestinal health and also affect distal organs, and there is extensive evidence that the influence of diet on health may be due, in part, to changes in the gut microbiome [170,171,172,173]. The effects of coffee on the gastrointestinal system have recently been reviewed, and it is clear that coffee modulates the composition of intestinal microbiota and microbial metabolites [55]. Moreover, in mouse models, caffeine-induced sleep restriction affects the composition of the gut microbiome and fecal metabolites [174]. For example, the nondigestible polysaccharides in coffee are rapidly metabolized to short-chain fatty acids in the gut, and this results in increased levels of *Bacteroides*/*Prevotella* species [175,176]. Another study also reported higher levels of *Bacteroides-Prevotella-Porphyromonas* in high consumers of coffee [177], and this was also accompanied by lower levels of lipoperoxidation and the increased microbial production of short-chain fatty acids that are chemoprotective in the gut [178]. It was also observed in a human dietary study that a high fiber/coffee (no red meat) diet improved insulin sensitivity in type 2 diabetes and decreased the expression of the pro-inflammatory marker interleukin-18 [179]. It was reported that *Prevotella* induced IL-18 and other inflammatory cytokines in the colon of SPF mice, whereas inflammation (DSS), induced colonic inflammation, and cytokine production were inhibited by *Prevotella*, and this was consistent with the overall beneficial effects of *Prevotella* [180], which is enhanced in coffee drinkers. The results of several other studies indicate that coffee induces bacterial species and their metabolites that are known to have beneficial effects [181,182,183]. However, due to the variability in the diet-induced changes of the intestinal microbes and their metabolites [172,184], the role of these coffee-induced effects on changes in microbial populations and microbial metabolites is not yet fully understood.

## 8. Summary

Coffee is not only the most consumed beverage worldwide, but it joins the Mediterranean diet as being among those dietary components that extend life, protect against neurological and liver diseases, and protect against the diseases of other organs. There is also an association between higher coffee consumption and overall anti-inflammatory effects and protection against some cancers, whereby coffee acts as both a chemopreventive and chemotherapeutic agent. The mechanisms of action of coffee are dependent on the effects of its constituents, including chlorogenic acids, polyphenolics, terpenoids, alkaloids, and other phytochemicals. Caffeine may contribute to some coffee-induced responses, but there are studies showing similar health benefits in individuals consuming caffeinated or decaffeinated coffee. There is evidence that the antioxidant activity of coffee, which activates Nrf2, may be an important mechanism of action. However, since Nrf2 exhibits both health-protective and drug-resistant activities, other cell context-dependent factors may also be important. There is also evidence that the protective effects of coffee in the gut and decreased colon cancer risk may be due to its activity as an AhR ligand. Moreover, some of the components of coffee bind the orphan nuclear receptor NR4A1 to the interactions with this receptor, and as of yet, unidentified receptors may also be important. Overall, these mechanisms, in concert with possible epigenetic pathways and the modulation of gut microbiota/microbial metabolites, contribute to the health benefits of higher coffee consumption, and this suggests that clinical applications of coffee extracts, particularly for treating some cancers, should be considered.

## Figures and Tables

**Figure 1 ijms-24-02706-f001:**
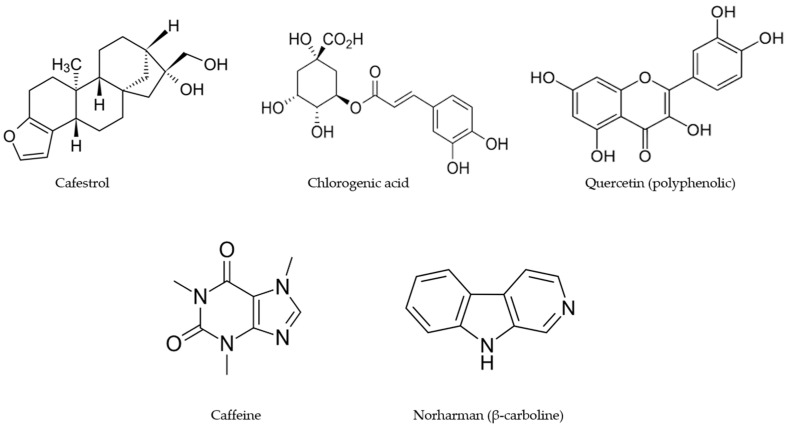
Structures of individual compounds in roasted coffee after extraction with hot water.

**Figure 2 ijms-24-02706-f002:**
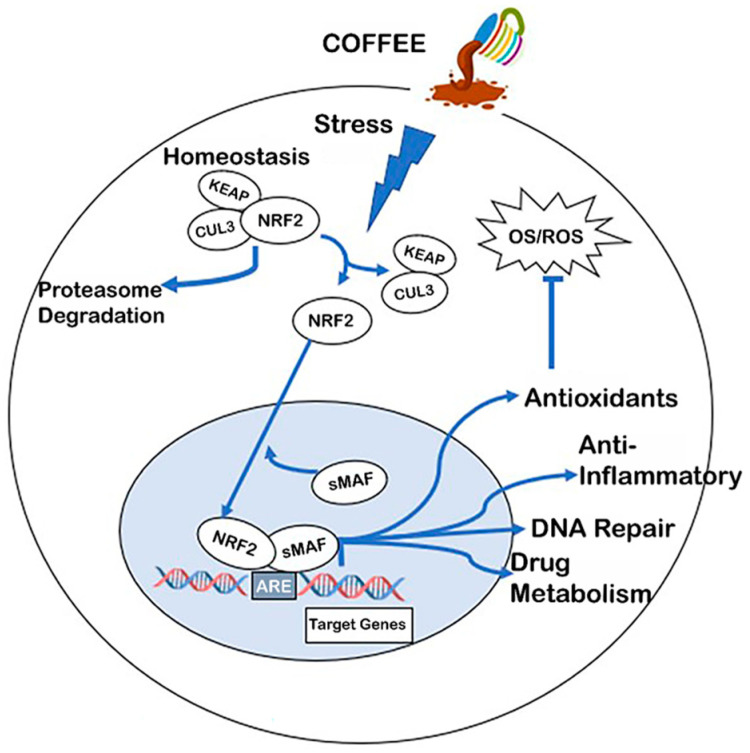
Development of intracellular oxidative stress is attenuated by coffee-induced stress. Coffee-induced stress results in the dissociation of NRF2 from KEAP and the subsequent translocation of NRF2 into the nucleus where the NRF2-sMAF complex binds cis-acting AREs to induce NRF2 regulated genes/pathways, including the antioxidant enzymes glutathione peroxidase (GPx), superoxide dismutase (SOD), quinone oxidoreductase 1 (NQO1), glutathione reductase (GR), harm oxygenase 1 (HO-1), and several thioreductases family members [104,105,106,107].

**Figure 3 ijms-24-02706-f003:**
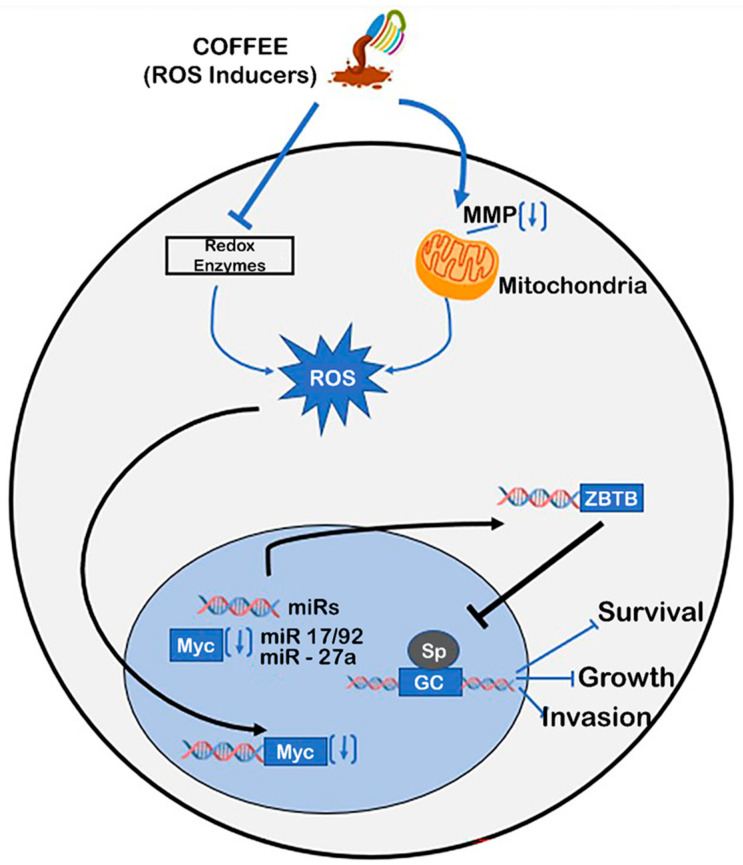
Coffee extracts induce ROS. Treatment of cancer cells with ROS inducers, such as chlorogenic acids and quercetin, inhibits redox enzymes and decreases mitochondrial membrane potential (MMP) to induce ROS, which downregulates cMyc + cMyc-regulated microRNAs (miRs: miR 17-92/27a). This results in the induction of ZBTB genes (ZBTB4, ZBTB10) + inhibition of pro-oncogenic Sp1/Sp3/Sp4-regulated genes/pathways [127,128,129,130,131,132,133].

**Figure 4 ijms-24-02706-f004:**
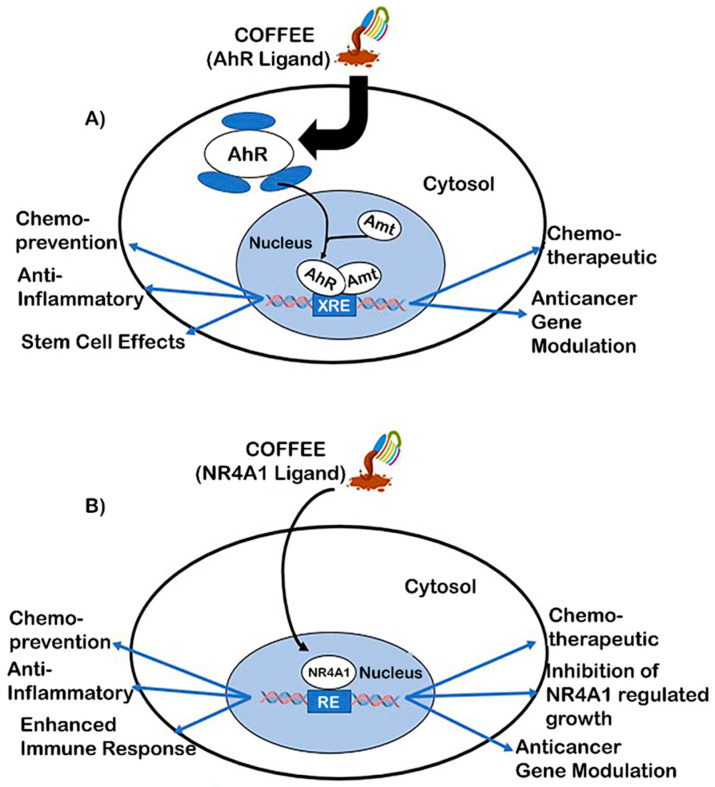
Coffee receptor-mediated responses: coffee extracts that bind AhR (**A**) or NR4A1 (**B**) may activate age- and cell-context-dependent chemopreventive and/or chemotherapeutic responses [156,157,158,159,160,161].

## Data Availability

Data available in a publicly accessible repository.

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
