# Peer review of "Health Benefits of Coffee Consumption for Cancer and Other Diseases and Mechanisms of Action"

_ijms, 2023, doi:10.3390/ijms24032706_

Round 1

Reviewer 1 Report

In the review entitled “Health Benefits of Coffee Consumption for Cancer And Other Diseases,” The authors analyzed the recent literature, from 2020, on coffee consumption, focusing mainly on its effects on cancer and citing only its impact on neurological and metabolic diseases and the difference between sex, which requires more attention.

1. The topic is not original or relevant to the field in its current form.
The authors should more accurately analyze the subject of their review, enrich the text, and add more detail about the effects of coffee consumption.

2. A review paper provides the reader with information about studies within the field's topic and explains the results of some already published research that argues for them. In the present form, the review needs to be expanded on some points. See next point.

3. The Authors should expand the text of the review, which is very concise. In particular, they have to improve the part regarding the effect of coffee consumption on neurological and metabolic disease and the difference between the sexes. In this regard, I can refer to some papers that have not been mentioned, for instance: "doi: 10.3390/nu12103080"; doi.org/10.1038/s41598-020-80302-4; 10.3390/ijms232314837.

4. The resolution of the Figures 2,3 and 4 should be improved as well as the legend of them.

To be suitable for publication, the revision required changes must be made.

Author Response

Thank you for taking the time to review our manuscript. Please find below our response to your reviewer comment.

1. Initially the review was to be focused primarily on cancer however this does not do justice to the overall health benefits of coffee. Therefore, we have increased the scope of the review as suggested and also have emphasized the "mechanisms of action" component of the review, which are also a major focus of the review (see changed title).

2. & 3.  As indicated above we have expanded the review as suggested by the Reviewer and have included the new papers.

4. The resolution of the Figures and the Figure captions have been improved.

Reviewer 2 Report

Thank you for the very informative and comprehensive review. I have these comments:

- For the Gastrointestinal tract (page 2), it seems that there is no association between coffee consumption and decreased risk of cancer in GIT. In this regard, please refer to this article https://www.ncbi.nlm.nih.gov/pmc/articles/PMC4635778/ which is a meta-analysis study of long term consumption of coffee and gastric cancer. Please cite it. It must be useful. 

- For the coffee consumption and risk of prostate cancer, the authors can depend on the results of the meta-analysis (#61) which suggests that a higher intake of coffee may be associated with a lower risk of prostate cancer. Please highlight this result without the paper. 

- For figure 1, for structure norharman, please add its common name b-carboline. Also, this figure seems to be confusing for the readers. Coffee contain several compounds not only shown in the figure, so, please add a text that these compounds are major compounds or even add the other compounds reported in coffee. Also, quercetin is a flavonoid. It is better to nominate it as a polyphenolic. 

- The resolution of Figure 2, 3 and 4 should be improved and also the text should be corrected. I see some red lines in the text from the word. Please improve it. 

- The authors wrote "thin-layer chromatographic analysis showed that multiple compounds which differ in polarities are responsible for the AhR activity in coffee." Please explain, it is known that TLC analysis is a technique which reveal the different compounds in the extract. How it could be correlated to the AhR activity in coffee as it is mentioned in the text?. Please explain. 

- There is almost 8 pages references while the text is about 10 pages, so please try to enrich your paper with texts. The review must be comprehensive and the readers should refer to it if they want to get a brief summary on the link of Coffee consumption and cancer, so please try to enrich the paper with more texts from the papers. Just be careful of plagiarism.

- Some typographical errors should be corrected. 

Author Response

Thank you for taking the time to review our manuscript. Please find below our response to your reviewer comment.

1. This material has now been incorporated into the review.

2. The prostate cancer results (#61) have now been highlighted.

3. The changes suggested for Figure 1 have now been made.

4. The resolution of Figures 2-4 and the text have been improved.

5. An explanation of the TLC results and their significance has been improved.

6. The text has now been expanded and typographical errors have been corrected.

Round 2

Reviewer 1 Report

The manuscript has improved since the first time I reviewed it. However, several unsettled issues are still necessary to address before publication is accepted.

1)The authors stated: “Moreover, one study also reports expression of higher plasma levels of several biomarkers of key metabolic and inflammatory pathways that are protective including c-reactive protein, high molecular weight adiponectin and 17β-estradiol”, and they refer to reference #33, which is incorrect. It pertains to metabolic disease, but not about that c-reactive protein, high molecular weight adiponectin, and estradiol 17B are protective.

- And about c-reactive protein, it is known that elevated c-reactive protein levels are associated with the onset and extent of activated inflammation. Therefore, the Authors should check and correct the reference and/or the sentence.

2) To better explain what the authors mean by the effect of coffee and NRF2, for example, they could add, in the text of the manuscript and in the legend of figure 2, that NRF2 is the regulator of the transcription of antioxidant genes such as GPx, SOD, HO- 1, GST.

- The legend of figure 2 should be more explanatory. See point 2

-As written in the previous revision, the Authors were too succinct in the text of the review. Still, they should detail the assertions they wrote.

e.g. “non-digestible polysaccharides in coffee are rapidly metabolized to short chain fatty acids in the gut and this results in increased levels of Bacteriodes/Prevotella species”.

And what is the consequence of this?? See the paper: 10.1038/s41385-020-0296-4

Author Response

We thank you for taking the time to review our manuscript, below please find our responses to your comments and suggestions:

1)The authors stated: “Moreover, one study also reports expression of higher plasma levels of several biomarkers of key metabolic and inflammatory pathways that are protective including c-reactive protein, high molecular weight adiponectin and 17β-estradiol”, and they refer to reference #33, which is incorrect. It pertains to metabolic disease, but not about that c-reactive protein, high molecular weight adiponectin, and estradiol 17B are protective.

- And about c-reactive protein, it is known that elevated c-reactive protein levels are associated with the onset and extent of activated inflammation. Therefore, the Authors should check and correct the reference and/or the sentence.

- Author Response: We apologize for the incorrect reference and have revised the text accordingly.

2) To better explain what the authors mean by the effect of coffee and NRF2, for example, they could add, in the text of the manuscript and in the legend of figure 2, that NRF2 is the regulator of the transcription of antioxidant genes such as GPx, SOD, HO- 1, GST.

- The legend of figure 2 should be more explanatory. See point 2

-As written in the previous revision, the Authors were too succinct in the text of the review. Still, they should detail the assertions they wrote.

- Author Response: We have added the NRF2 genes in text and legend for Figure 2. In addition, we have amplified the text with more detail.

e.g. “non-digestible polysaccharides in coffee are rapidly metabolized to short chain fatty acids in the gut and this results in increased levels of Bacteriodes/Prevotella species”.

And what is the consequence of this?? See the paper: 10.1038/s41385-020-0296-4

- Author Response: We have amplified the text and addressed the Reviewer’s point and other reports on Prevotella species.